# Visual Memory for Robust Path Following

**Ashish Kumar**\*    **Saurabh Gupta**\*    **David Fouhey**    **Sergey Levine**    **Jitendra Malik**
University of California, Berkeley
ashish_kumar@berkeley.edu, {sgupta, dfouhey, svlevine, malik}@eecs.berkeley.edu

## Abstract

Humans routinely retrace paths in a novel environment both forwards and backwards despite uncertainty in their motion. This paper presents an approach for doing so. Given a demonstration of a path, a first network generates a path abstraction. Equipped with this abstraction, a second network observes the world and decides how to act to retrace the path under noisy actuation and a changing environment. The two networks are optimized end-to-end at training time. We evaluate the method in two realistic simulators, performing path following and homing under actuation noise and environmental changes. Our experiments show that our approach outperforms classical approaches and other learning based baselines.

## 1  Introduction

Consider the first morning of a conference in a city you have never been to. Rushing to the first talk, you might follow your phone's directions through a series of twists and turns to reach the venue. When you return later in the day, you can retrace your steps to your hotel relatively robustly, remembering to take a left turn at the bistro and keep straight past the coffee shop. The next day, you may probably only look at your phone to check your email. At first glance, this seems like a trivial ability. Humans routinely do this, for instance when a friend shows you the bathroom in their apartment or when you go to a room in a new building. On second glance though, it is an amazing ability since one never retraces one's steps exactly and the visual experience is constantly changing in fairly dramatic ways: people move their cars, shops open and close, and seasons change. This paper aims to replicate this ability to retrace paths (including reversals) in new environments with imperfect ability to replicate one's actions (actuation) as well as a changing world.

How might we solve this problem? One classical approach, common in robotics, would be to build a full 3D model of the world via SLAM, from building facades to the side-mirrors of cars, during the first pass; after this is done, path following reduces to localizing in the model and selecting the best action. For the task of navigation, this is simultaneously too much work – precisely reconstructing the facade is less important than recognizing it as "the bistro at which I turn left" – as well as too little work – the parts of the reconstruction that provide stable localization and the parts that do not are mixed together with no way to disentangle them.

Given these difficulties of classical approaches, a large number of learning-based approaches have sprung up to solve this and other related navigation tasks (*e.g.* [28, 27, 18, 35, 43, 30]). In these works, an agent directly predicts actions from image observations in an end-to-end fashion. However, unlike our setup of a single demonstration in a new environment, many of these setups require the agent to have a great deal of experience with the *test* environment via reward based supervision [27, 28, 43] or exhaustive human demonstrations [35]. Moreover, unlike most work on navigation in new environments [18], our setup poses the additional challenge of noisy actuation as well as a world that can change between the initial demonstration and path execution.

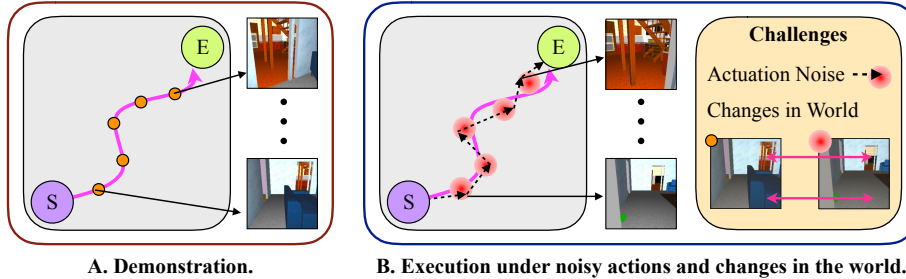

| A. Demonstration. | B. Execution under noisy actions and changes in the world. |

**Figure 1: Problem Setup:** Given images and associated actions from a traversed path (A), we want to retrace the path both forward and backward under actuation noise (*i.e.* uncertainty in movement) in a changing world (*e.g.* the chairs have been moved between the demonstration and execution) (B). This paper presents an end-to-end neural architecture for this task. We use the demonstration path to generate a path abstraction that is used by a learned controller to reliably convey the agent to the desired target location at execution time.

Our approach, which we describe in Section 3, consists of a module that learns to convert a series of observations of a path to an abstract representation, and a learned controller that *implicitly* localizes the agent along this abstracted path using the current observation and outputs actions that bring the agent to the desired goal location. We see a number of advantages to training the whole approach end-to-end on data in comparison to classical approaches. First, the learned model can use statistical regularities to make its performance more robust: for example, it can learn to count doors in a texture-less hallway rather than localize at each point and when it returns along a path, it can learn to look on the right for a table that was previously on the left. Second, by virtue of being learned entirely end-to-end, rather than being learned and designed piecemeal, the approach can automatically learn the features that are necessary for the task at hand. As a concrete demonstration, we evaluate a homing task in which the agent retraces a path in reverse; the network learns to produce features necessary for solving this task without explicitly designing any wide-baseline features or proxy tasks.

We evaluate our approach on multiple datasets in Section 4 in a series of experiments that aim to probe to what extent we can learn to retrace a path under noisy actuation and in a changing world. We compare to a variety of classical and learned alternate approaches, and outperform them. In particular, our experimental results show the value of end-to-end learning the entire path-following process.

## 2    Related Work

In this work, we study the problem of retracing a path under noisy actuation and a changing world. This touches on classical work in robotics and recent works learning-based efforts.

One classic approach to solving the path retracing problem (also referred to as Visual Teach and Repeat [15]) is to chain together core robotics primitives of mapping, localizing, and planning: on the initial demonstration, the agent builds a 3D map; during subsequent navigation attempts, the agent localizes itself in the map and generates actions accordingly. Each problem has been studied extensively in robotics, typically focusing on geometric solutions, *i.e.* using metric maps and locations. For example, mapping has been often studied as the classic simultaneous localization and mapping (SLAM) problem [12–14, 19] in which one builds a 3D map of the world. Localization is often framed as recovering camera pose with respect to a global or local 3D map [9, 10] or by performing visual odometry [29, 42] to obtain a metric displacement from a start position. Finally, planning is often done assuming direct access to a noiseless map and often a noiseless agent location [6, 22, 25].

The distinction from this line of work is that our approach is learned end-to-end. This means that intermediate representations are automatically learned, rich non-geometric information can be incorporated, and modules are optimized jointly for the end-goal. Obviously, researchers are aware of the limitations of purely geometric and pipelined approaches, and have developed extensions: explicitly incorporating semantics into SLAM [4], adding margins in planning to account for uncertainty [3], and using learning for sub-sets of the SLAM problem [5, 8, 21, 23, 32, 36, 40]. The strength of our proposed method is that these strategies are learned automatically and are specified by what is needed empirically to navigate an environment as opposed to human intuition.

We are not the first to recognize the potential for end-to-end task-driven navigation. In recent years there has been a flurry of work in this area. A lot of this work focuses on the shorter time-

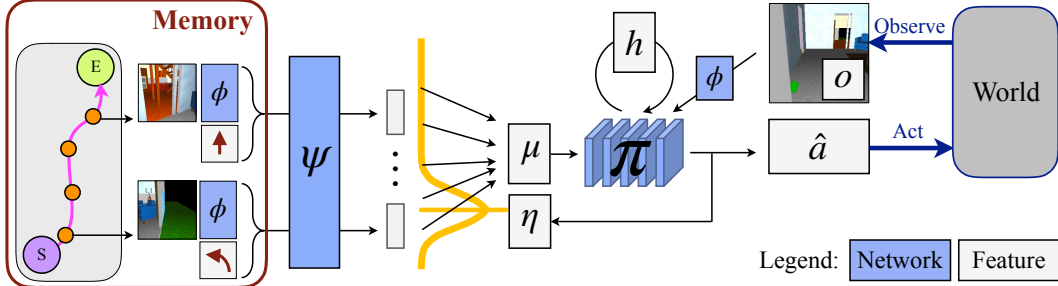

**Figure 2: Proposed Approach:** As input, our model takes a sequence of images (processed by $\phi$) and actions at these images $\mathbf{a}(\mathbf{p})_j$. It abstracts this sequence into a sequence of memories. A second, recurrent, network $\pi$ uses this sequence of memories to emit actions that retrace the path. At each time-step, $\pi$ reads in: (a) the sequence, softly attending to the relevant part at past time-step via $\eta$; (b) the current observation $O$ from the world (also processed by $\phi$); as well as its recurrent hidden state $h$. As output, $\pi$ updates the attention location $\eta$ and its hidden state, and emits action $\hat{a}$ that the agent executes in the world.

scale task of collision avoidance, *e.g.* how do I move through the door without bumping into it [11, 16, 17, 20, 33, 34]. In contrast, our work focuses on the longer time-scale problem of path following *e.g.* how do I get back to my office from the coffee machine. At this time-scale, some works have framed the problem of navigation as learning to reach different goals in a fixed training environment [27, 28], designing policies that can directly act in new test environments [18, 24, 30, 41], self-supervised learning for reaching goals in a well-traversed environment [35] or by following a demonstration [31]. Our work is more similar to this last line of work [31, 35]. It is distinguished, however, by its lack of explicit localization as well as its navigation under *actuation noise* and *changing environments*. This noise is a crucial distinction because noiseless execution of actions and an unchanged environment means that memorization alone is sufficient. We also study the task of homing, where no direct demonstration images are available.

## 3 Robust Path Following

**Problem Setup.** Consider an agent that we are trying to train to operate in a new environment $E$ that it has never been in before. Let us assume that the agent's state at time $t$ is represented by $s_t$, and that the agent has some primitive set of *stochastic* actions $\mathcal{A}$ that it can execute. Executing an action $a \in \mathcal{A}$, takes the agent to state $s_{t+1}$ via a stochastic transition function $f$ *i.e.* $s_{t+1}$ is sampled randomly from the distribution $f(s_t, a, E)$. We assume that the agent is equipped with a first person RGB camera to obtain visual observations of the environment $I = \rho(E, s)$, where the function $\rho$ renders an environment $E$ from agent's current location specified by $s$. We do not tackle the problem of higher-level planning and abstract away low-level motor control to focus on the problem of robust path following. Our approach can be used with classical (or even learned) approaches for path planning and low-level motor control.

Suppose we move this agent around in this new environment along a path $\mathbf{p}$. Given such a traversal, we want the agent to be able to reliably re-trace the path $\mathbf{p}$ or to follow a related path such as the reversed path (denoted $\tilde{\mathbf{p}}$). We want to be able to do this in situations where the agent has noisy actuation and sensing, and the environment changes (to $E'$) between the demonstration of $\mathbf{p}$ and when the agent is tested to autonomously traverse $\mathbf{p}$ (or $\tilde{\mathbf{p}}$). Note that, the only sensory information that is available to the agent as it is trying to traverse the path $\mathbf{p}$ (or $\tilde{\mathbf{p}}$) in the environment $E$ (or $E'$) is via $\rho$. It does not have access to the ground truth state of the system $s_t$, but only RGB image observations of the environment from that state.

Our goal is to learn a policy $\Pi$ that will achieve this. As input, $\Pi$ assumes access to the: path $\mathbf{p}$; actions $\mathbf{a}(\mathbf{p})$ taken while executing the path; visual observations along the path $\mathbf{I} = \{I_1 \ldots I_J : I_j = \rho(E, \mathbf{p}_j)\}$; as well as the current visual observation $O$. As output, $\Pi$ predicts actions from $\mathcal{A}$ that successfully and efficiently convey the agent to the destination $\mathbf{p}_J$, *i.e.* $\hat{a} = \Pi(\mathbf{p}, \mathbf{I}, O)$. This action $\hat{a}$ is executed in the stochastic environment; the agent obtains the new visual observation at the next state; and this process is repeated for a fixed number of time steps.

We now describe the policy function $\Pi$. Intuitively, our policy function $\Pi$ uses the observation $O$ to implicitly localize the agent with respect to its memory of the path. Given this localization, the policy

reads out relevant information (such as relative pose and actions) and uses this in context of $O$ to take an action that conveys the agent to the desired target location. The entire policy is implemented using neural network modules that are differentiable and learned end-to-end using training data for the task of efficiently going to the desired goal location.

We first describe the basic architecture for $\Pi$ where we want to retrace the same path $\mathbf{p}$ that was demonstrated and then describe the extension to retrace a related path $\tilde{\mathbf{p}}$. We call our proposed policy $\Pi$ as Robust Path Following policy and denote it by RPF.

### 3.1 Learned Controller for Robust Path Following

The policy $\Pi(\mathbf{p}, \mathbf{I}, O)$ is realized as follows. We first use $\mathbf{p}$ and $\mathbf{I}$ to compute a path description $M(\mathbf{p})$ that captures the local information needed to follow the path. $M(\mathbf{p})$ is a sequence of tuples consisting of features of the reference image, associated reference position and the associated reference action for each step $j$ in the trajectory, or

$$M(\mathbf{p}) \;=\; \{(\mathbf{a}(\mathbf{p})_j, \phi(I_j)) : j \in [1 \ldots J]\}. \tag{1}$$

We use this path description with a learned controller that takes as input the current image observation $O$ to output actions to follow the path under noisy actuation. We represent the policy $\pi$ with a recurrent neural network that iterates over the path description $M(\mathbf{p})$ as the agent moves through the environment. This iteration is implemented using attention that traverses over the path description. At each step, the path signature is read into $\mu_t$ with differentiable soft attention centered at $\eta_t$:

$$\mu_t \;=\; \sum_j \psi\left(M(\mathbf{p})_j\right) e^{-|\eta_t - j|}. \tag{2}$$

The recurrent function $\pi$ with state $h_t$ is implemented as:

$$h_{t+1}, b, \hat{a} = \pi(h_t, \mu_t, \phi(O)) \text{ and } \eta_{t+1} = \eta_t + \sigma(b). \tag{3}$$

As input, it takes the internal state, $h_t$, attended path signature $\mu_t$ and featurized image observation $\phi(O)$. In return, it gives a new state $h_{t+1}$, pointer increment $b$, and action $\hat{a}$ that the agent should execute. This pointer increment is passed through an increment function $\sigma$, and added to $\eta_t$ to yield the new pointer $\eta_{t+1}$. Given the agent moves by one step per action in expectation, we use $1 + \tanh$ as the increment function $\sigma$. We set $\eta_1 = 1$ and $h_1 = \mathbf{0}$.

Note that we factor out the controller from the path description. This factorization of the environment and goal specific information into a path description that is separate from the policy lets us learn a *single* policy $\pi$ that can do different things in different environments with different path descriptions without requiring any re-training or adaptation. The policy can then also be thought of as a robust parameterized goal-oriented closed-loop controller.

### 3.2 Feature Synthesis for Following Related Paths

So far our approach can only repeat the paths that we have already taken, but our approach can be extended to follow paths $\tilde{\mathbf{p}}$ that are related to but not the same as the path $\mathbf{p}$ that was demonstrated. We do this by *synthesizing* features for the path $\tilde{\mathbf{p}}$ using whatever information is available for the demonstrated path $\mathbf{p}$. We synthesize features $\hat{\phi}(\tilde{\mathbf{p}}_j)$ for location $\tilde{\mathbf{p}}_j$ using observed features $\phi(I_i)$ from location $\mathbf{p}_i$ as follows:

$$\omega_{i,j} = \Omega\left((\phi(I_i), \delta(\mathbf{p}_i, \tilde{\mathbf{p}}_j))\right) \text{ and } \hat{\phi}(\tilde{\mathbf{p}}_j) = \Sigma_g\left(\omega_{1,j}, \omega_{2,j}, \ldots, \omega_{N,j}\right). \tag{4}$$

Here, the function $\delta$ computes the relative pose of image $I_i$ with respect to the desired synthesis location $\tilde{\mathbf{p}}_j$. $\phi$ computes the representation for image $I_i$ through a CNN followed by two fully connected layers. $\Omega$ fuses the relative pose with the representation for the image to obtain the contribution $\omega_{i,j}$ of image $I_i$ towards representation at location $\tilde{\mathbf{p}}_j$. These contributions $\omega_{i,j}$ from different images are accumulated through a weighted addition by function $\Sigma_g$ to obtain the synthesized feature $\hat{\phi}(\tilde{\mathbf{p}}_j)$ at location $\tilde{\mathbf{p}}_j$. The path description $M(\tilde{\mathbf{p}})$ can then be obtained as a collection of tuples $(\mathbf{a}(\tilde{\mathbf{p}})_j, \hat{\phi}(\tilde{\mathbf{p}}_j))$.

**Implementation Details.** We now describe the particular architecture that we use throughout the paper. $\phi$ is a 5 layer Convolutional Network with [32, 64, 128, 256, 512] filters respectively. Each conv layer is followed by a max-pooling. $\psi$ and $\Omega$ are fully connected networks consisting of two layers; $\pi$ is implemented using GRUs. We train the entire network from scratch in an end-to-end manner using Adam optimizer for 120000 iterations, where each episode is 40 steps long.

# 4 Experiments

This paper studies the task of retracing a route in a *new* environment (either forwards or backwards) under noisy actuation and a changing world, given a demonstration. Our experiments evaluate: a) to what extent can we solve this task, b) what is the role of visual memories, and c) how our proposed solution compares to classical geometry-based and other learning-based solutions. Crucially, we further study how our policies perform on settings outside of what they were explicitly trained on.

## 4.1 Experimental setup

**Simulators.** We use two simulation environments that permit rendering from arbitrary viewpoints and allow separation of held-out environments for testing. The first simulator is based on real world scans from *Stanford Building Parser Dataset* [2] (SBPD) and the *Matterport 3D Dataset* [7](MP3D). These scans have been used to study navigation tasks in [18], and we adapt their publicly available simulation code. We use splits that ensure that the testing environment comes from an entirely different building than the training or validation environments. The second simulation environment is based on *SUNCG* [39]. SUNCG consists of synthetic indoor environments with manually created room and furniture layouts that have corresponding textured meshes [39]. Because these environments are graphics codes, SUNCG permits evaluation of the effect of environmental changes. In particular, objects can be removed without inducing artifacts that a network will pick up on. Once again, splits ensure no overlap between training and testing environments.

**Agent Actions.** We assume that the agent has 4 macro-actions, stay in place, rotate left or right by $\theta$ $(= 30°)$, and move forward $x$ units $(= 40cm)$.

**Noise Model.** Our work studies path retracing both with actuation noise (*i.e.* the outcome of actions is stochastic) and a changing world (the world changes between the demonstration and autonomous operation of the agent). *Actuation Noise:* In both environments, when the agent outputs the rotation actions it actually rotates by $\sim N_{trunc}(\theta, 57.3°, 0.2|\theta|)$. Here, $N_{trunc}(\mu, \sigma, \delta)$ refers to a normal distribution with mean $\mu$ and standard deviation $\sigma$ that is truncated to $(\mu - \delta, \mu + \delta)$. When the agent executes a move forward action it rotates by $\sim N_{trunc}(0, 57.3°, 0.2|\theta|)$ and then translates by $\sim N_{trunc}(x, 5cm, 0.2x)$. The factor of $0.2$ controls the amount of noise in actuation and we vary it in our experiments. *World Changes:* World changes are studied in the SUNCG environment. Demonstrations are provided in an environment with objects removed uniformly with a probability of $r$ $(= .5)$. The task is to get to the desired target location in presence of even fewer objects ($r$ is .1, or .3) or additional objects ($r$ is .7, or .9).

**Tasks.** Given a path $\mathbf{p}$ from $\mathbf{p}_0$ to $\mathbf{p}_T$, we consider the two tasks of *trajectory following i.e.* going from $\mathbf{p}_0$ to $\mathbf{p}_T$ as well as *homing i.e.* going from $\mathbf{p}_T$ to $\mathbf{p}_0$. We evaluate these tasks under a variety of noise conditions and environments.

**Evaluation Criteria.** We characterize the success of the agent by measuring how close it gets to the goal location. We analyze each approach over 500 trials in a novel environment (not used for training). We report three metrics: a) *Success Rate* (reaching within 2 steps or $10\%$ of the initial distance to goal, whichever is larger), b) *SPL* (Success weighted by normalized inverse Path Length as described in [1]), and c) *Median Normalized Distance* to goal at end of episode.

**Model Training.** We use imitation learning to train our policies. Although the agent never has access to its true location as it traverses the environment, the true location is available in the simulator. This is used to compute a set of 'good actions' that will convey the agent to the desired target location. This set of good actions are ones that lead to a larger reduction in the distance to goal when compared to forward action. We optimize the policy to minimize the negative log of the sum of probabilities of all good actions at each time step.

## 4.2 Baselines

We compare with a number of baselines that represent either classical approaches or test the importance of various components of our system.

**Open Loop.** We repeat the reference actions (or their reverse for homing). Under perfect actuation this would achieve perfect performance. With actuation noise, this baseline is a measure of the hardness of the task and tests to what extent learning to act under actuation noise is necessary.

**Table 1:** Performance over 500 trials on the test set (SBPD `area4`) in base settings for the Following and Homing tasks. We report Success Rate, SPL and Median Normalized Distance. See text for details.

| | Open Loop | Visual Servoing | 3D Recons. + Localize | RPF (no visual mem.) | RPF |
|---|---|---|---|---|---|
| **Following** | | | | | |
| Success Rate ↑ | 0.216 | 0.318 | 0.826 | 0.782 | **0.866** |
| SPL ↑ | 0.191 | 0.293 | **0.766** | 0.648 | 0.726 |
| Median Normalized Distance ↓ | 0.257 | 0.204 | 0.086 | 0.068 | **0.056** |
| **Homing** | | | | | |
| Success Rate ↑ | 0.216 | – | 0.000 | 0.780 | **0.866** |
| SPL ↑ | 0.191 | – | 0.000 | 0.644 | **0.740** |
| Median Normalized Distance ↓ | 0.257 | – | 0.853 | 0.062 | **0.056** |

**Visual Servoing.** For each action, we compute the $L2$ distance between SIFT feature matches of target reference image (initially set to first reference image) and image expected after executing that action (we obtain this image by *virtually* executing this step in the simulator). The policy actually executes the action that has the lowest distance. To decide when to increment the target image, we check the ground truth proximity to the next reference image. We stop when the agent reaches the last target image.

**3D Reconstruction and Localization.** We use the publicly available COLMAP package [37, 38] that implements a variety of geometric mapping and localization algorithms. Note that these geometry-based methods require high-resolution images at high frame rates. Thus, we sample high-resolution images ($1024 \times 1024$ *vs.* $224 \times 224$ for our policies) at $5\times$ the frame rate ($145$ *vs.* $30$ images for our policy for a trajectory of length 30) along the reference trajectory along with ground truth poses. This ensures that reconstruction via SIFT key-point matching [26] and bundle adjustment always succeeds. Given this reconstruction, the agent localizes itself by registering SIFT key-points on the current image with the 3D reconstruction derived from the reference images. It then estimates free space by marking a small region around each point on the reference trajectory as free. Given this inferred free space and localization, it executes the action that most efficiently conveys it to the goal location. If the localization fails, it rotates left at each step until localization succeeds (or the episode ends).

**RPF with No Visual Memory.** We also compare to a policy without any visual memory. We do this by only using the action and pose as part of the memory $M(\mathbf{p})$ in Eq. 1 *i.e.* removing $\phi(I_j)$. We retain the rest of the architecture of RPF. Note that this is a competitive comparison as the policy can learn to replay actions meaningfully in context of current images from the environment.

All learned policies are trained in *base settings*: the noise level is at $20\%$, reference trajectories are of length 30, the policy is executed for 40 time steps, and these reference trajectories are sampled to be far from obstacles. Figure 3 and Table 1 present our experimental results. We report the success rate, SPL, and median normalized distance and compare against the baselines described above. In addition, we also show a bootstrapped $95\%$ confidence interval for each plot.

## 4.3 Results

**Experiments on Matterport Data.** We first study the following and homing tasks in static environments. As the environment does not need to change, we do these experiments on realistic Matterport data. In particular, we train policies on 4 floors from SBPD and 6 buildings from MP3D. All policies are tested on `area4` from SBPD which is from an altogether different building than the 4 floors used for training. We found adding visually diverse data from the MP3D dataset was crucial for good performance as our models are trained entirely from scratch.

**Following Task.** Table 1 (top) presents results for the trajectory following task. Open loop replay only succeeds 22% of the time. This is because actuation noise compounds over time and causes both drift and collisions. Visual feedback through servoing improves success rate to 32%, though it is limited as it only localizes against a single image at a time. This is addressed by '3D Reconstruction + Localization' which uses all reference images for localization (by reconstructing the environment), and achieves a success rate of 83%. RPF achieves a higher success rate of 87%. RPF without visual memories succeeds 78% of the time. This is a striking result. Just the sequence of actions, when replayed intelligently through a learned controller, leads to compelling path tracing performance. That is, sequences of actions can robustly be interpreted in context of current image observations without any explicit 3D reconstruction or localization. Of course, additionally using visual memories from the demonstrated path improves performance.

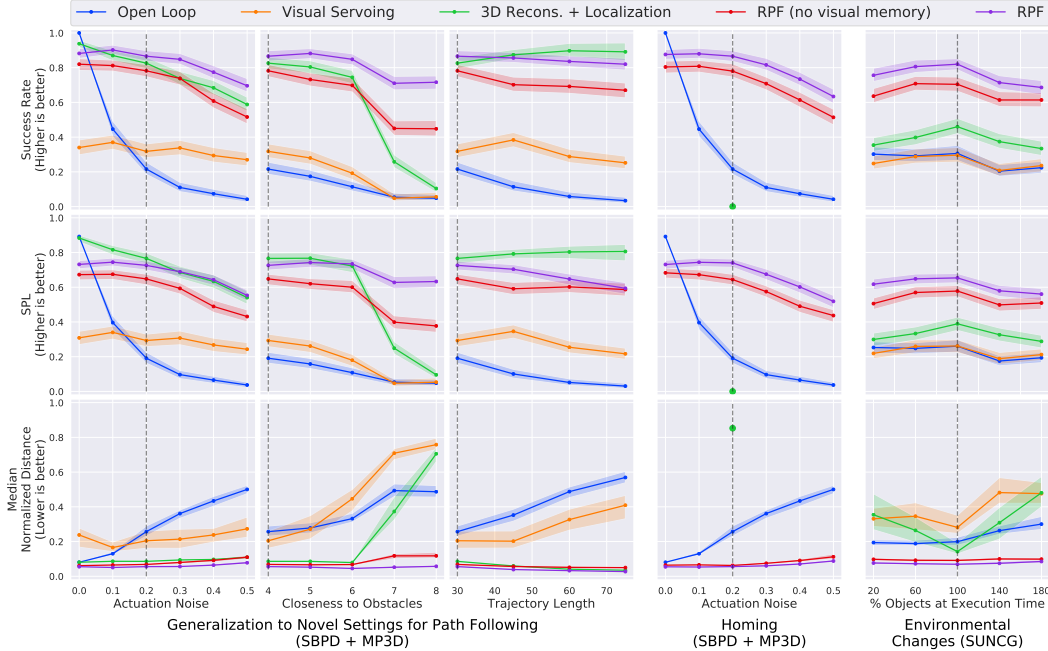

**Figure 3: Generalization Performance:** Rows show different metrics, and columns show different test settings. Vertical dotted lines mark 'base settings' that the policies were trained on. We show generalization of RPF to (far left) actuation noise levels, (center left) obstacle distance, and (center) path length for path following, and (center right) actuation noise for homing. The far right plot shows performance under environmental changes.

**Homing Task.** Table 1 (bottom) shows results for the homing task on Matterport data. We set up the homing task by sampling the same trajectories but simply providing images from a 180° rotated camera at each reference location. Thus open loop replay and RPF without visual memories perform about the same as before. Note that this scenario is impossible for visual servoing as there are no direct images to servo to. While, in principle 3D Reconstruction and Localization can tackle this scenario, it performs poorly (0% success, 0.85 median normalized distance) as SIFT features don't match well across large baselines. In comparison, RPF that learns to speculate features still performs equally well at 87%. Note that this is better than RPF without visual memories, demonstrating that our feature prediction technique is able to extract meaningful signal from related images.

**Testing on Out-of Train Settings.** We have shown so far that our trained policies outperform appropriate baselines for the task when tested on novel environments. However, we are still training and testing on the same settings, such as noise level, trajectory length and distribution of trajectories. We next test how well our learned policies work when we test them on settings they have not been trained on. We do this by testing the policies trained in the 'Base Setting', in different settings in novel environments. We explore three novel settings: a) *Actuation Noise*: we vary the noise level of the environment, b) *Harder Trajectories*: we pick trajectories that are closer to obstacles and require careful maneuverability, and c) *Path Length*: we increase the length of the trajectories to follow. We do not retrain our policies for these settings, and simply execute the policy learned in the 'Base Setting' on these additional settings.

Figure 3 (left 3 columns) presents the results. Rows plots the different metrics (success rate, SPL, and median normalized distance), and columns plot different test conditions. Plots in the first column show performance as a function of actuation noise. Policies trained at 20% noise are tested under noise varying between 0% and 50%. Open loop performs perfectly with no noise, but its performance rapidly degrades as noise increases. In comparison, RPF performance degrades gracefully. Plots in the second column show performance as we move to harder trajectories that are sampled to be closer to obstacles (as we move from the left to the right on the plot). RPF degrades much more gracefully than the 3D reconstruction based method. The third column shows performance as a function of trajectory length. RPF policies generalize reasonably well to even 3× longer trajectories than were seen during training. Column 4 shows similar plots of performance variation as a function of the actuation noise for the homing task. Finally, Figure 4 shows multiple rollouts from our RPF policy and contrasts them with purely open loop rollouts.

| Following (Matterport) | Homing (Matterport) | Changing Env (SUNCG) |

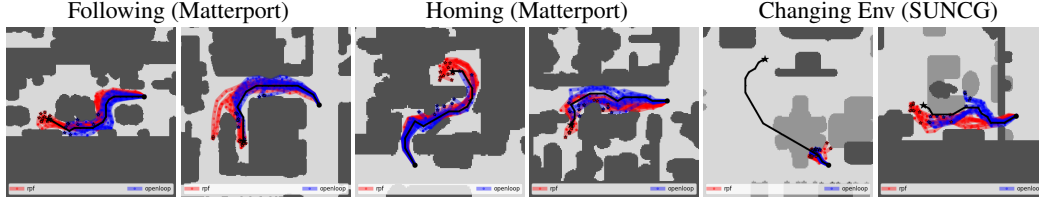

**Figure 4: Trajectory Visualizations:** We visualize sample success and failure trajectory in top view for the three different scenarios. We show multiple roll-outs by resampling multiple times from the noise model. Note that these top views are shown here only for visualization, the policy does not receive them and operates purely based on the first person views. RPF trajectories are in red, and open loop ones are in blue. **Following and Homing**: RPF trajectories more tightly follow the reference trajectory, while open loop roll-outs spread out and collide with nearby obstacles. Overall, RPF gets to the goal more reliably, but fails when it drifts too far from the reference trajectory. **Execution in Changing Envirnonment in SUNCG:** Our approach is able to go around obstacles, but fails if a very large deviation is required.

**Experiments on SUNCG.** We next report experiments on SUNCG for the base setting. We trained on 48 houses from the House3D training set and report performance on 12 entirely disjoint houses from the test set. We plot the metrics in Figure 3 (last column). The base setting (shown with the dotted vertical line) here corresponds to when there are 100% objects at execution time *i.e.* the environment is the same between when the reference trajectory was provided, and when the agent has to execute the trajectory. As RPF learns features, it can better adapt to change in visual imagery in the synthetic dataset as compared to feature-based geometric methods. Once again, RPF was trained in base settings.

**Robustness to Environmental Changes.** Figure 3 (last column) also plots performance as the environment changes. Points to the left of the dotted line correspond to the setting where objects are removed from the environment, while points to the right correspond to when objects are added into the environment. In both these regimes, RPF continues to perform well. In contrast, performance for the 3D reconstruction and localization method degrades sharply (see median normalized distance plot) as the environment at execution time deviates from one at demonstration time. This is expected and known of geometry based methods that do not cope well with changes in the environment. Moreover, the agent doesn't just need to be robust to visual changes between reference images and the current observations, but it must also exhibit local going-around-behavior as the reference trajectory may go through the newly added obstacles. While RPF is able to do so when there is a minor detour, it fails when a much larger detour is required as shown in Figure 4 (right).

**Ablations and Comparisons to Other Learning Methods.** Figure 5 reports success rate for the path retracing task for other learning methods: a) *Trained Nearest Neighbors*: we use similarity between current observation with reference images to vote for the action, and b) *GRU*: a standard recurrent model without the proposed sequential memory. We also compare to ablations of RPF: a) *RPF without Recurrence*: $\pi$ is implemented with a feed-forward network as opposed to a recurrent function, and b) *RPF with a Constant Increment* instead of the learned increment

function. RPF does better than these other learning methods, and our choice of modeling $\pi$ as a recurrent function, and learning how to increment the pointer are important. We also investigated other choices for implementing the increment function $\sigma$. Increment functions with a small spread around 1 worked better than other choices.

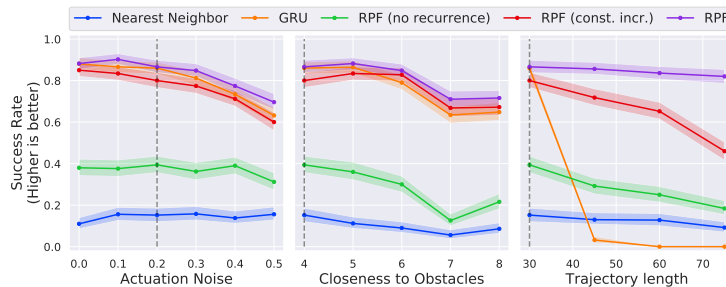

**Figure 5: Other Comparisons:** We report success rate for the path following task on SBPD + MP3D for other learning based methods. See text for details.

**Discussion.** In this paper, we studied the task of following paths under noisy actuation and changing environments. We operationalized insights from classical robotics with learning-based models and designed neural architectures that implicitly tackle localization to output actions to convey the robot directly to the target location. We made thorough comparisons with geometry-based classical methods and reasonable learning baselines, and demonstrated the effectiveness of our proposed approach.

## Footnotes

\*Equal contribution. Project website with videos: https://ashishkumar1993.github.io/rpf/.

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
