[Reviews · NeurIPS 2018]

Reviewer 1



This paper proposes to address the interesting problem of how to retrace a path in a novel environment. For which a two-stage end to end learning approach is developed. The first stage involves an abstract of a path, while the second is for the human or agent to retrace the path in the presence of action noise and a changing world. The presentation is clear in general. The problem is interesting and the proposed algorithm is reasonable and novel. Empirical simulations demonstrate the good performance of the proposed approach. On the other hand, there is a bit lacking in terms of real experiments on practical situations. Some additional comments: line 23, pp.1:solvethis -> solve this line 69, pp.2:The ... distinction this line of work and our method is ...-> The ... distinction between existing work and our method is ... line 70: pp.2:...learned and incorporate nongeometric information and modules are jointly optimized... -> ...learned, nongeometric information is incorporated, and modules are jointly optimized...

Reviewer 2



This paper considers the problem of retrace the trajectory from noisy visual observations. The proposed approach firstly takes a sequence of images of the path and encode as sequential memories. In the online setting, a neural network is able to attend to the past memory and take actions accordingly. The algorithm has been verified over two interesting 3D indoor scene dataset. Pros: * The paper is very well-written. Introduction is clearly motivated, novelty and relationship with previous works have been discussed in a very modest tone, approach is very clear and figures are well illustrated. I really appreciate the author’s efforts on writing and enjoy reading this paper. * Proposed algorithm is well designed. I especially like how the proposed approach could be related with the mapping, localization and motion planning procedure in traditional robotics. * The author considered challenging cases where path is not exactly the same and changes present in the world. * It shows an interesting, promising, yet unsolved direction to NIPS community, which I believe would draw attentions in short future. Cons: * The proposed method is limited to work only with very related paths (with limited perturbed noise or reversed trajectory). And it’s difficult to see the potential of generalize to a very different path using the current feature synthesis algorithm. Human indeed has this capability with very limited exploration. * Failure cases were not discussed in detail. There are half of the cases failed. Could you analyze more of those cases and point out potential future directions to authors? * Baselines (especially classic SLAMs) are not favoured under current setting. * Recurrent settings might be restricted to short paths, e.g. 30 steps in paper’s setting. It is also unclear to me what information hidden states take to policy function. Could you report performance with a non-recurrent policy pi(mu_t, Phi(O_t))? Some detailed comments: It seems SLAM baseline suffers a lot in this simple scenario, in which world changes are not very significant. This is quite surprising to me given the capacity of COLMAP. Thus I would like to know more about the author’s experiment on the SLAM baseline to ensure a proper comparison: * Are you using incremental bundle adjustment or full bundle adjustment setting in COLMAP? * Do you give BA solver your true camera intrinsic matrix? * How many frames you used per each path? Still 30? And action between each step is also pure rotation with 30 degree or 40cm forward? Note that both settings are not realistic in real-world applications (images could be received at much higher rate) and could make SLAM based approaches unfavoured. * Have you actually managed to estimate accurate camera pose and reconstruct dense point cloud with good quality using COLMAP under your setting? Do you think localization failures are due to poor mapping or poor SIFT matching? * In the path following stage, why do only do SIFT matching against the points only, rather than the full incremental BA pipeline (SIFT matching plus non-linear reprojection error minimization) to estimate the new image of the camera pose? Robotics community does have a lot of online localization algorithm with loop closure detection capability, e.g. Fabmap. These works should be definitely considered to compare as stronger geometric baselines.

Reviewer 3



The authors address the topic of path following, namely how to train a deep network model that is able to remember a path in a novel environment and later retrace it. The classical approach to this would be simultaneous localization and mapping, however, the authors argue this is a needlessly complex task. In the proposed method, the agent does not explicitly measure its location or try to localize, it uses only image inputs to remember the path. The deep network consists of two sub-networks. The first, consisting of a convolutional network to encode the images and two fully-connected layers, serves to create an abstract representation of the shown path. The second, a recurrent neural network that uses the encoded path and reconstructs it emitting a series of actions. The complete network is trained end-to-end through reinforcement learning where any action leading to reduction of the distance to goal is rewarded. The paper addresses an interesting question with a new approach. The presentation is relatively clear. I would suggest acceptance. Remarks: - I would expect the implementation details section to come earlier in the paper, i was lacking the definition of phi, psi, pi, omega. - please check the usage of \pi vs \Pi - The pointer increments are passed through a sigmoid, does this mean that the attention cannot change by more than a single time point? This seems rather restrictive, is it needed? - L78-90 the writing style of this paragraph makes it hard to understand - please check the sentence "Note that..." (starting on L107), it does not make sense to me UPDATE: Thanks for the clear authors response. I support accepting this paper.